# Highly Selective Gas-Phase Catalytic Hydrogenation of Acetone to Isopropyl Alcohol

**Abdulrahman A. Al-Rabiah** [1,*] , **Ismail Boz** [2] , **Vagif M. Akhmedov** [3] , **Mohamed Mokhtar M. Mostafa** [4] **and Abdulaziz A. Bagabas** [5,*]

1   Chemical Engineering Department, King Saud University, P.O. Box 800, Riyadh 11421, Saudi Arabia
2   Department of Chemical Engineering, Istanbul University—Cerrahpasa, Avcilar, 34320 Istanbul, Turkey
3   Institute of Catalysis and Inorganic Chemistry, National Academy of Sciences of Azerbaijan, H. Javid Av. 113, AZ1143 Baku, Azerbaijan
4   Department of Chemistry, King Abdulaziz University (KAU), P.O. Box 80203, Jeddah 21589, Saudi Arabia
5   National Petrochemical Technology Center (NPTC), Materials Science Research Institute (MSRI), King Abdulaziz City for Science and Technology (KACST), P.O. Box 6086, Riyadh 11442, Saudi Arabia
*   Correspondence: arabiah@ksu.edu.sa (A.A.A.-R.); abagbas@kacst.edu.sa (A.A.B.); Tel.: +966-11-4676844 (A.A.A.-R.); +966-11-4813790 (A.A.B.)

**Abstract:** Current industrial synthesis procedures of isopropyl alcohol (IPA), by the direct or indirect hydration of propylene in the gas or liquid phase, suffer from the low conversion of propylene, the requirement for high pressure, and the harmfulness to the environment. In this context, we report a single-step, gas-phase process for the green synthesis of IPA via acetone hydrogenation, in a fixed-bed reactor, under ambient pressure and within a temperature range of 100–350 °C. Composite catalysts with various ratios of ruthenium nanoparticles supported on activated charcoal and *nano*-zinc oxide ($n$-Ru/AC/$n$-ZnO) were used. Catalytic activity and selectivity were functions of $n$-Ru/AC/$n$-ZnO loading ratios, reaction temperature, and the hydrogen to acetone molar ratio. The composite catalysts were characterized by X-ray powder diffraction (XRPD), transmission electron microscopy (TEM), hydrogen temperature-programmed reduction ($H_2$-TPR) analysis, and nitrogen physisorption. High yields of IPA were obtained over 3$n$-Ru/AC/2$n$-ZnO) catalyst, which showed the highest selectivity of 98.7% toward isopropyl alcohol and acetone conversion of 96.0% under a hydrogen to acetone mole ratio of 1.5 at 100 °C. Reaction rates, calculated from the model equation, were in reasonable agreement with those measured experimentally. The apparent activation energy ($E_a$) value for acetone hydrogenation was found to be 17.2 kJ/mol. This study proved that immobilized Ru catalysts were potential superior catalysts for the selective hydrogenation of acetone to IPA in exceptionally mild green synthesis conditions.

**Keywords:** acetone hydrogenation; isopropyl alcohol; ruthenium-activated charcoal; zinc oxide; heterogeneous catalysis; kinetic modeling

## 1. Introduction

Nowadays, there is an abundance of acetone (dimethyl ketone, DMK), mainly because of dominant phenol production and its availability as a by-product of bio-butanol process and biomass conversion technologies [1]. However, acetone consumption is far less than that being produced. Major uses of acetone include the synthesis of bisphenol A, methyl isobutyl ketone, methyl acrylate, chemical heat pumps, and some minor hydrogen storage schemes. One of the prominent ways for utilizing acetone is the direct catalytic hydrogenation to IPA [2–4]. It is well-known that acetone condensation leads to a myriad of compounds, such as mesityl oxide (MO), methyl isobutyl ketone (MIBK), methyl isobutyl carbinol (MIBC), isophorone (IP), diacetone alcohol (DA), and some by-products, such as propane, propene, and isobutene [5].

Hence, the one-step synthesis of the more valuable product of IPA requires the development of a highly selective, active heterogeneous catalyst [6,7]. Direct catalytic hydrogenation is economically feasible and attractive. In selective hydrogenation reactions, heterogeneous ruthenium-based catalysts have emerged potentially [7]. In this respect, there are numerous studies where some showed high yields under relatively harsh conditions [8,9], and some exhibited low yields under relatively mild conditions [2,9]. All the industrially accepted manufacturing procedures lack selectivity and moderate operating conditions [9,10] in comparison to the technique used in our current study.

The current situation, as of 2020, is quite havoc due to the coronavirus pandemic. In addition, the demand for acetone is close to saturation because the production at major downstream bisphenol A (BPA) and methyl methacrylate (MMA) units is stable. Furthermore, acetone prices dropped to USD 400 per metric ton (MT) as of March 2019. During the same time period, isopropyl alcohol prices in the USA were raised to USD 1662 per MT, FOB [Texas], in the quarter ending in March [11]. This historical price comparison shows us that there is still a great additional value for the production of isopropanol from acetone with novel technologies by using moderate operating conditions.

As is well-known in this art of using metallic catalysts, the catalytic performance depends on various factors such as nanoparticle size, nature of support, ease of regeneration, and reaction conditions [4]. The prior art has previously suggested that ruthenium hydrogenation catalysts could be reactivated by reducing the catalyst. Besides $SiO_2$ [12], other non-reducible supports such as carbon nanotubes (CNT) [13], and MgO [14] were used as supports of ruthenium and were found to contribute to changes in the catalytic activity and selectivity in the hydrogenation of acetone and aliphatic aldehydes. Because the selective hydrogenation of aldehydes and ketones is known to be a reaction sensitive to metal– support interactions, it was stimulating to test the properties of Pt/ZnO catalysts for this reaction, knowing that ZnO showed promoting effects in several studies. A marked decrease in the catalytic activity with a significant increase in selectivity was observed in butadiene hydrogenation to olefins when the Pd/ZnO catalyst [15] was reduced above 423 K. In this context, our paper is devoted to the hydrogenation of acetone in the gas-phase, at ambient pressure, on Ru deposited on activated charcoal and composited with ZnO to improve the adsorption of acetone, under various operating conditions of temperature, flow rate, and $H_2$/acetone mole ratio. Thus, we aimed to combine the good hydrogenation ability of ruthenium with the adsorption capabilities of ZnO.

This study also aims to achieve the direct catalytic hydrogenation and high yields of IPA under more benign conditions. Here, we reported a systematic study relevant to the synthesis of IPA at high yields under 100–350 °C and atmospheric pressure, over a series of composite catalysts of *n*-Ru/Ac and *n*-zinc oxide.

## 2. Results and Discussion

### 2.1. Characterization Results

The major characteristics of the five *n*-Ru/AC:*n*-ZnO composite catalysts are listed in Table 1. The *n*-ZnO had a surface area of 37 m$^2$/g, far smaller than that of *n*-Ru/AC (average 300 m$^2$/g). Ruthenium metal content by weight increases in the following order: CAT-V > CAT-IV > CAT-III > CAT-II > CAT-I.

The total BET surface areas of ruthenium-containing composite catalysts averaged around 300 m$^2$/g, whereas metallic surface areas changed from 24.4 m$^2$/g for CAT-II, to its maximum of 30.7 m$^2$/g for CAT-III, and its minimum of 22.4 m$^2$/g for CAT-V. Metallic surface areas did not follow the trends in BET surface areas.

Crystalline phase analyses were carried out by XRD. Figure 1 shows the XRD patterns of the four exemplary supported catalyst samples. The XRD pattern of CAT-V had peaks at 2θ = 26.3°, 45.2°, and 53.9°, which were assigned to the activated carbon support of *n*-Ru/Ac. Ruthenium was also observed at 2θ = 44.2° (JCPDS card 6-0663), though it was barely visible. The XRD pattern of Cat IV showed the presence of a graphitic carbon phase, and the overriding ZnO phase was dominant over the metallic Ru phase. Major phases

detected were wurtzite, amorphous carbon, and ruthenium albeit in small peaks. XRD patterns of ruthenium over activated carbon mixed with *n*-ZnO did not show sharp peaks, which could not be easily assigned to the ruthenium metallic phase. This finding suggested that ruthenium particle sizes were smaller than 3–4 nm, as confirmed by high-resolution TEM images.

**Table 1.** Textural properties and characteristics of investigated catalysts [6].

| Catalyst Code | Catalyst | Ru Loading (wt.%) | $S_{BET}$ (m²/g) | $S_{Ru}$ (m²/g) | Crystallite Size (nm) | Major Phase |
|---|---|---|---|---|---|---|
| CAT-I | n-ZnO | 0.0 | 37 | – | 20 | Wurtzite |
| CAT-II | 1n-Ru/AC:2n-ZnO | 1.67 | 300 | 24.4 | 23 | Wurtzite |
| CAT-III | 1n-Ru/AC:1n-ZnO | 2.5 | 289 | 30.7 | 25 | Wurtzite |
| CAT-IV | 3n-Ru/AC:2n-ZnO | 3.0 | 278 | 25.7 | 30 | Wurtzite |
| CAT-V | n-Ru/AC | 5.0 | 327 | 22.4 | – | amorphous |

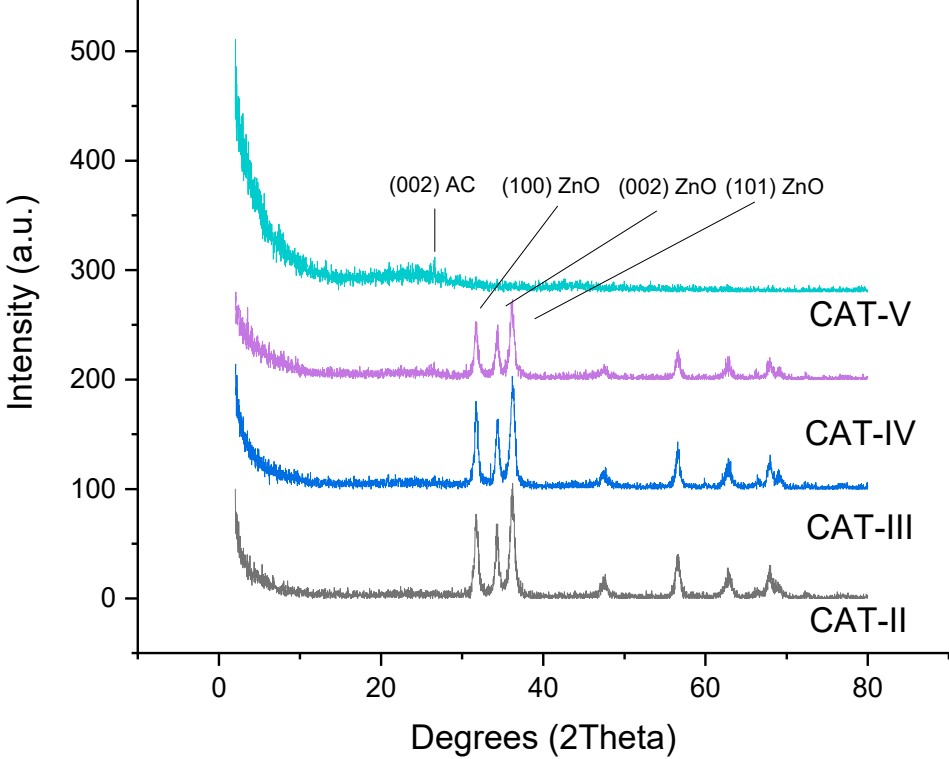

**Figure 1.** X-ray powder diffractions of some catalysts, CAT-V: n-Ru/AC and CAT-IV: 3n-Ru/AC:2n-ZnO [6].

Figure 2a,b show complementary pairs of fresh crystalline *n*-ZnO particles. There were crystalline particles with sizes between 20 and 30 nm rather than homogeneous sizes. Figure 2c,d show the amorphous nature of activated carbon and small ruthenium particles in the range of 3–4 nm.

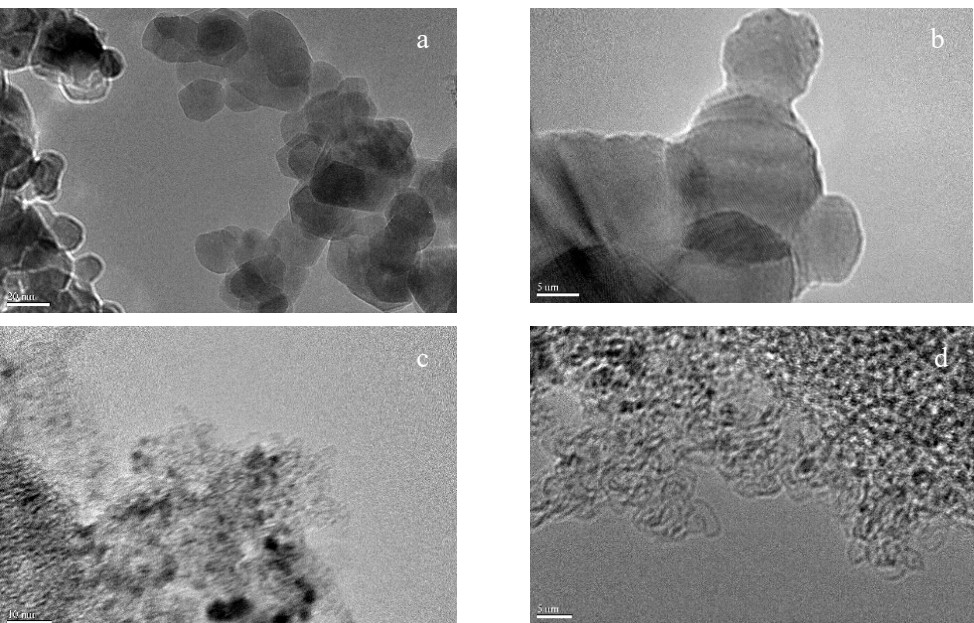

**Figure 2.** TEM images of unused CAT-I: n-ZnO (**a**,**b**) and CAT-V: n-Ru/AC catalyst (**c**,**d**).

## 2.2. Effect of Temperature

The effect of temperature on the catalytic performance of the mentioned composite catalysts was investigated between 100 °C and 350 °C at a fixed $H_2$/DMK mole ratio of 4.0 and ambient pressure. Figure 3 shows the variation of acetone conversion percent and product selectivity percent over the composite catalyst and time-on-stream (TOS = 1 h) at 100, 150, 200, 250, 300, and 350 °C. As shown in Figure 3, decreasing temperature led to a considerable increase in acetone conversion and IPA selectivity. The highest acetone conversion (83%) was observed over CAT-IV at 100 °C. CAT-IV also showed 99.6% selectivity toward IPA and almost zero selectivity toward MIBK. However, over CAT-IV at 350 °C, acetone conversion was 21% and the selectivity toward IPA was 20%.

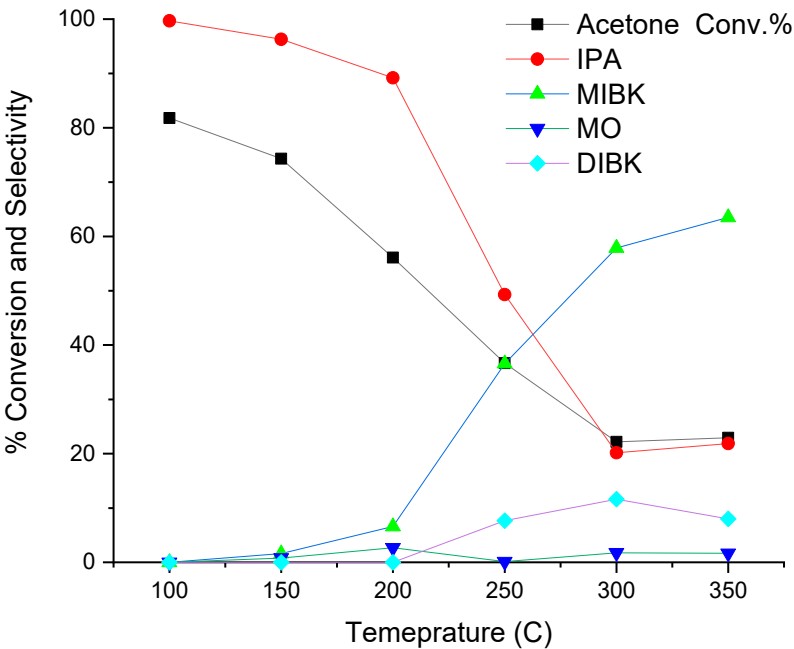

**Figure 3.** Variation in the conversion of acetone and selectivity at reaction conditions: 0.25 g CAT-IV catalyst, $H_2$/DMK mol ratio = 4, TOS = 1-h.

These observations indicated that addition and condensation reactions were favored over higher temperatures, while the direct hydrogenation reaction of acetone was favored at lower temperatures. Shutkina [12] found dissimilar trends for the selective hydrogenation of acetone at 2.5 MPa over an 11% $Cu/SiO_2$ catalyst. They found that acetone conversion increased from 10% to 91% in the range of 130–170 °C, while the selectivity towards IPA decreased from 100% to 97.5%. The decrease in acetone conversion at high temperatures was explained by thermodynamic limitations. Although high operating pressures should favor the product side, the results were contrary to this thermodynamic explanation. However, this discrepancy was due to the high partial pressures of hydrogen rather than thermodynamic control. On the other hand, at higher partial pressures of hydrogen and temperatures, ruthenium-based catalysts exhibited a decrease in acetone conversion. This finding was in line with their results, in which they showed that $Ru/SiO_2$ was practically inactive at elevated temperatures. High operating temperatures led to lower adsorption of acetone over ZnO support, which was the main cause of the lower conversion of acetone, as it was proven previously by the absence of adsorbed acetone above 698 K by the vibrational spectroscopy [13]. Moreover, the catalyst identity plays a key role in acetone conversion and in directing the reaction toward either MIBK or IPA. In heterogeneous catalytic carbonyl hydrogenation, a general mechanism, referred to as C=O activation, is provided by the electropositive species such as $H^+$ [14], $Ru^{4+}$ [15], $Zn^{x+}$ [16] on the catalyst surface. In the C=O activation, these electropositive species can coordinate to the oxygen atom in the C=O bond and polarize the carbonyl group, facilitating the attack of C=O by the positively hydrogen species. The generation of dissociative hydrogen is a facile reaction on highly active metal surfaces, such as ruthenium, while selective, non-competitive adsorption of acetone on zinc oxide surfaces occurs below 698 K [13]. Therefore, we are required to have additional surfaces for the adsorption of acetone in competition with hydrogen.

The molecular adsorption of acetone occurs because the carbonyl oxygen atom contacts either to the metal surface [17] or, here in our catalyst, to nonpolar zinc oxide sites [13]. The dissociated hydrogen atoms on the metal surface then attack the adsorbed acetone, where hydrogenation of the enolated carbonyl-carbon atom first occurs via isopropoxy intermediates (partially hydrogenated intermediates), followed by consecutive hydrogenation of these intermediates to isopropanol.

*2.3. Effect of Hydrogen to Acetone Mole Ratios*

The effect of the hydrogen to acetone mole ratio is listed in Table 2. Reaction conditions were selected to favor IPA synthesis. As listed in Table 2, the higher mole ratio of hydrogen to acetone led to a lower conversion of acetone. However, in a broad range of conversion from 35% to 96%, selectivity toward IPA did not change and was optimized at 99%. If the adsorption processes of reactants ($H_2$, acetone) and isopropanol product on the Ru metal catalyst surface competed to a certain extent, the initial hydrogenation rate of acetone would decrease gradually with an increase in the hydrogen partial pressure, corresponding to the excess surface coverage of hydrogen with respect to the acetone. At higher hydrogen partial pressure, the rate would decrease owing to the higher surface coverage of hydrogen in comparison to acetone, resulting in lower reaction rates.

**Table 2.** Effect of hydrogen to acetone mole ratio on conversion and selectivity at 100 °C.

| $H_2$/DMK Mole Ratio | Conv., % | Selectivity, % | | | | | | |
|---|---|---|---|---|---|---|---|---|
| | | MIBK | DIBK | MO | M | IPA | DA | Others |
| 1.5 | 96.0 | 0.0 | trace | 0.0 | 0.0 | 98.7 | Trace | 1.14 |
| 3.0 | 93.9 | 0.0 | trace | 0.0 | 0.0 | 99.8 | Trace | 0.17 |
| 4.5 | 87.0 | 0.0 | trace | 0.0 | 0.0 | 99.6 | Trace | 0.31 |
| 6.0 | 35.4 | 0.0 | trace | 0.0 | 0.0 | 99.8 | Trace | 0.13 |

Reaction conditions: 0.25 g CAT-IV, temperature = 100 °C, time-on-stream = 1 h; (DIBK: diisobutyl ketone; M: mesitylene).

## 2.4. Effect of Space Velocity

The effect of space velocity was studied at 250 °C, while keeping the acetone flow rate constant at 0.01 mL/min and changing the hydrogen flow rate. The results are shown in Figure 4. Within the experimental range studied here, the conversion did not change considerably. However, the selectivity toward IPA increased as the hydrogen flow rate increased. At a low hydrogen flow rate or low hydrogen partial pressures, other products such as MIBK and DIBK due to the parallel thermally driven reactions were observed, while at a high hydrogen flow rate or high hydrogen partial pressures (high coverage), the only plausible activity was toward IPA with suppression of addition and condensation reactions. This finding is in line with the kinetic modeling, where dissociative adsorption of hydrogen is the rate-determining step. Similar findings were observed over metallic copper supported on alumina [13], where Basu and Pradhan explained this phenomenon due to the competitive nature of multifunctional sites responsible for direct hydrogenation and MIBK synthesis.

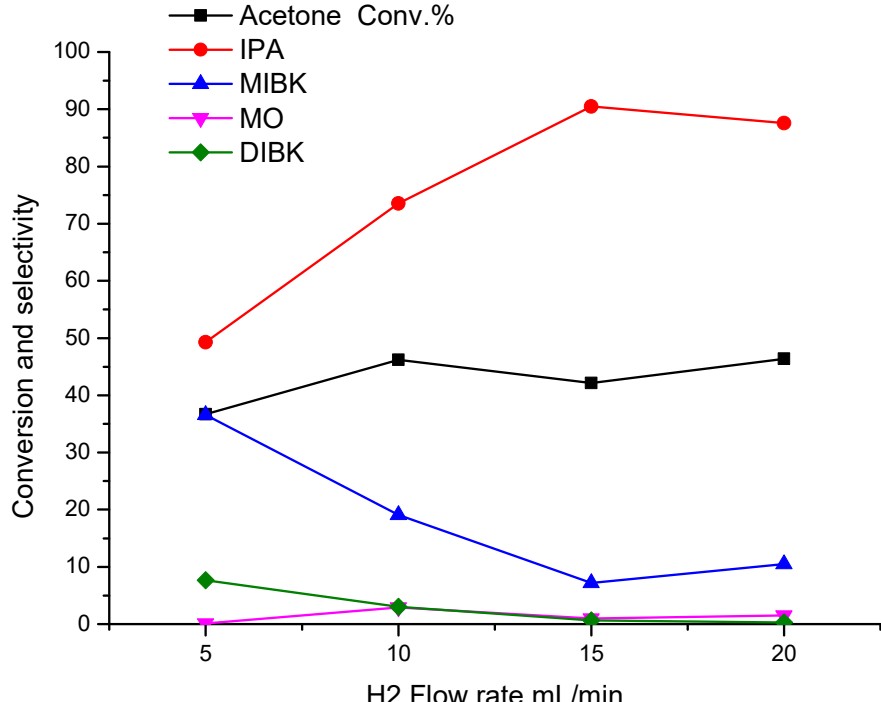

**Figure 4.** Variations of conversion and selectivity over CAT-IV at reaction conditions: 0.25 g catalyst, t = 250 °C, acetone flow rate = 0.01 mL/min, TOS = one hour.

Dissociation of molecular hydrogen on ruthenium is a rather slow process [14], and low coordinated defect sites may be the preferential sites for a direct dissociative adsorption pathway on ruthenium nanoparticles [15]. At low temperatures, the catalytically active ruthenium surface is normally covered to a large extent with hydrogen. The surface coverage remains incomplete under reaction conditions, even at elevated pressures. Thus, at 100 bar, a coverage θ of ca. 85% was calculated at room temperature, whereby it decreased to ca. 70% at increasing temperature (500 °C) [16].

## 2.5. Effect of Time on Stream

Figure 5 shows the changes in the reaction conversion and selectivity toward some selected major products with time-on-stream (TOS) over the CAT-IV catalytic system. There were some minor fluctuations observed in conversion: after one hour of TOS, acetone conversion was 46.4%, and it was then increased a fraction to 48.69% (the maximum). However, after five hours of TOS, the conversion was 44.97%. The formation of IPA was 87.56% selectivity at the beginning and slightly declined to 83.12% after five hours. On the

other hand, the selectivity toward MO and DIBK began increasing until it reached 2.62% and 1.95% after five hours, respectively. After five hours, MIBK selectivity increased from 10.53% to 11.74%. Cu-based acetone hydrogenation catalysts [17] were found to be highly structure-sensitive and deactivated by time-on-stream. The deactivation was likely due to the coverage by inactive carbon. In our experiments, we did not observe such drastic deactivation, which was probably due to the high activity and selectivity of Ru-based catalysts and good dispersion activated carbon support. The presence of *n*-ZnO somehow prevented the reactions leading to the condensation and hence barred the formation of heavier adducts, leading to the carbonaceous species, especially at higher temperatures [18]. This tentative explanation was experimentally proven from IPA decomposition studies over zinc oxide catalysts [19]. Cracking species with a molecular weight smaller than IPA begun to appear at around 244 °C to 326 °C. This finding proved that both acetone and IPA begin to decompose at rather elevated temperatures.

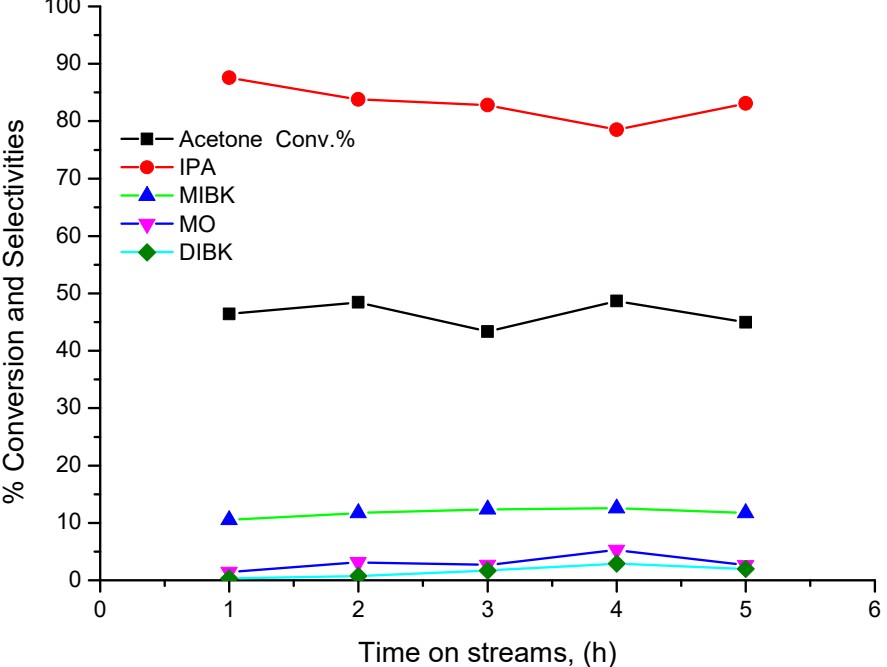

**Figure 5.** Product selectivities and acetone conversion over CAT-IV catalyst. Reaction conditions: 0.25 g catalyst, t = 250 °C, $H_2$/acetone mole ratio = 6:1.

Although some minor changes were observed, we have concluded that the CAT-IV catalyst, 3*n*-Ru/AC:2*n*-ZnO, was rather stable even under such high temperatures while it maintained its high selectivity toward IPA synthesis. The separate experimental results at lower temperatures provided a clear indication that the CAT-IV catalyst was a very efficient, cost-effective, selective hydrogenation catalyst for acetone under ambient pressure.

### 2.6. Effect of Ruthenium Loading

The conversion and selectivity as a function of Ru-loading are shown in Figure 6. Acetone conversion increased with increasing Ru-loading up to 3.0 wt.% and then decreased with increasing the Ru-loading to 5.0 wt.%. The selectivity toward MIBK increased with increasing the Ru-loading up to 2.5 wt.% and then it reduced with increasing the Ru-loading. The observed maximum MIBK selectivity, at 2.5 wt.% Ru-loading, could be attributed to the presence of balanced multi-functional sites (hydrogenation and condensation). On the other hand, increasing Ru-loading resulted in increasing the IPA selectivity except at Ru-loading of 2.5 wt.%. This observation indicated that high Ru-loading favored the direct hydrogenation of the acetone carbonyl group. The multifunctional composite catalyst with 1*n*-Ru/AC:0*n*-ZnO (CAT-V) exhibited the highest IPA selectivity with a lower conversion.

It appeared that the metallic sites were very essential for the formation of IPA. A maximum in the conversion of acetone was observed as ruthenium content increased over PtRu/C catalysts [8]. Similar to our findings, although the conversion showed some variations in acetone conversion, IPA selectivity was above 80% except for the *n*-ZnO (CAT-I) catalyst. Thus, it proved the necessity of the presence of metallic ruthenium and ruthenium oxide for high selectivity [4].

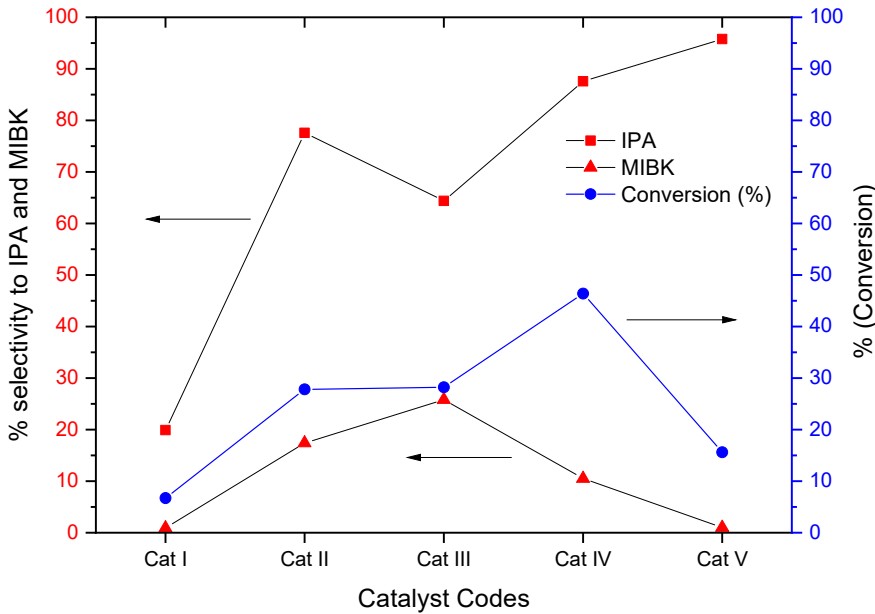

**Figure 6.** The conversion and selectivity as a function of Ru-loading. Reaction conditions: 0.25 g catalyst, t = 250 °C, $H_2$/acetone mole ratio = 6:1.

Crotonaldehyde hydrogenation to crotyl alcohol over Ru/ZnO catalysts [18] bore some similarities. As the ruthenium content increased from 0.5% Ru/ZnO to 5.0% Ru/ZnO, conversion increased gradually, while selectivity to crotyl alcohol (hydrogenation product) reached a maximum of over 3.0% Ru/ZnO catalyst and then decreased drastically. In line with our findings, ruthenium content presented a maximum for the yield of IPA. However, they attributed the high deactivation to the presence of chlorine and Lewis acid sites, which acted as detrimental additional steps. In our catalyst, we have not observed such deactivation due to the difference in the pyruvic oxime route for ZnO preparation. When the support exhibits strong acid character, such as heteropoly acids, it is well-known in the order of catalytic activity: Pt > Ru > Ni > Cu, over heteropoly acid support predominantly gives products, such as alkanes and alcohol dehydration species [20]. On the other hand, the use of basic supports, such as magnesium aluminum layered double oxides, resulted in product streams containing myriads of species: IPA, MIBK, MO, diisobutyl ketone (DIBK), and 3,3,5-trimethylcyclohexanone (TMCO), along with the changes in metallic content of the Pt/MgO catalysts [21]. The changes in the conversion and selectivity to IPA were a function of not only metallic content but also of the acid/base character of the support [2,15–17].

### 2.7. Kinetics Modeling

A tentative reaction scheme is shown below in Figure 7. In this scheme, we included only the compounds found in considerable concentrations, namely IPA and MIBK, though present in lesser quantities. Therefore, other minor compounds, especially those observed at high temperatures, were omitted. We used the data obtained between 100 °C and 350 °C. Figure 7 depicts the hydrogenation route to IPA and the condensation route to the MO, which was not observed due to the high catalytic activity of the catalysts tested for hydrogenation.

**Figure 7.** Reaction network scheme and possible pathways of acetone hydrogenation and condensation.

The kinetics data were collected under internal and external diffusion-free conditions. Because in the product distribution, the major compounds were composed of IPA and MIBK, a Langmuir–Hinshelwood–Hougen–Watson (LHHW) type kinetic model was selected to include these compounds [22]. The kinetic models of Basu and Pradhan [13] and also the models developed by Sen and Vannice [23] were applied after necessary data manipulations to avoid zero division in POLYMATH nonlinear regression analysis. Residuals of calculated and experimental data were minimized by the Levenberg–Marquardt algorithm, also known as the Damped least squares (DLS) method. From modeling the quality of kinetic data, we applied the models of the above-mentioned studies by using a limited set of experimental data points in which selectivity to IPA and repeatability of the experiments were of high quality. It was found that the model of Basu and Pradhan (Equation (1)), namely, dissociative adsorption of hydrogen, was assumed to be the rate-determining step.

$$- r_{acetone} = \frac{k\, P_{H_2}}{\left(1 + K\, P_{acetone}\right)^2} \tag{1}$$

where $k$ is the rate constant, $K$ is the equilibrium constant, $P_{H_2}$ and $P_{acetone}$ are the partial pressures of hydrogen and acetone, respectevily.

Although this model fits the kinetic data, it cannot be conclusively said that the actual mechanism follows the mechanism proposed, unless the model is backed up by spectroscopic and other means.

The goodness of fit was depicted in Figure 8. The correlation coefficient was found to be a mere 0.86. This finding was due to the very wide range of temperatures selected, whereas limited data, covering the partial pressures, were available for model testing.

In the open literature, there is an apparent discrepancy in the identification of the rate-determining step [22,23]. Because the number of reactants is two, namely, hydrogen and acetone, and because they are simple in molecular complexity, there is a consensus on the mechanistic steps leading to the IPA over metal catalysts. Yet, there is no solution to the nature of species on the surface of metallic catalysts by spectroscopic techniques.

They found that the surface reaction of adsorbed acetone and dissociatively adsorbed atomic hydrogen was the rate-determining step. Unlike their model-fitting results, dissociative adsorption of hydrogen was found to be the rate-determining step. From the Arrhenius plot as shown in Figure 9, activation energy was calculated to be 17.2 kJ/mol, whereas Basu and Pradhan [13] obtained 44.3 kJ/mol, and Sen and Vannice [23] found around 60 kJ/mol. Apparent activation energies varied between 11 and 14 kcal/mol over various copper-based catalysts which showed structure sensitivity [17].

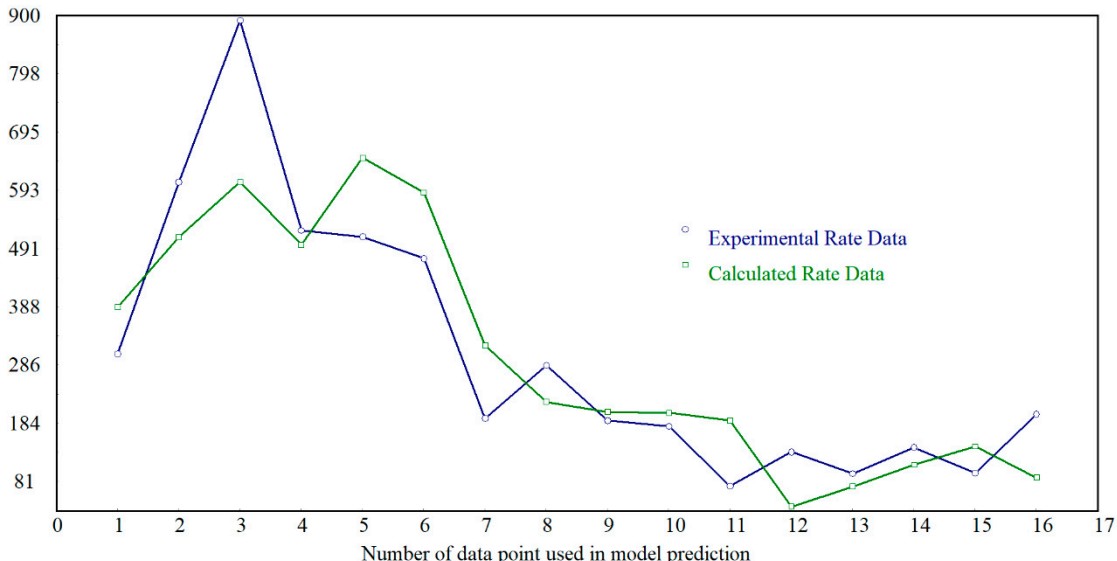

**Figure 8.** The goodness of fit data for the model Equation (1).

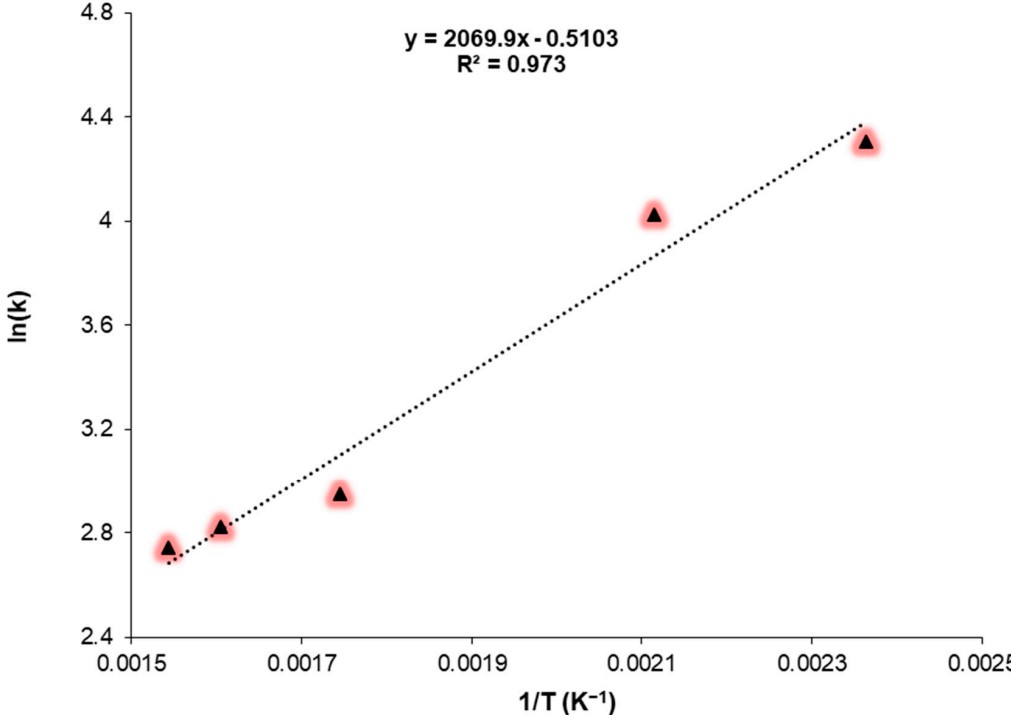

**Figure 9.** Arrhenius plot for IPA synthesis over CAT-IV.

Our catalyst exhibited much higher activity and selectivity toward IPA. These low activation energy findings are in line with kinetics data. We demonstrated that under moderate conditions (100 °C, 0.1 MPa $H_2$), it is possible to selectively and quantitatively hydrogenate acetone to highly valuable IPA (selectivity 99.8%).

Furthermore, an important feature of this reaction system is that *n*-ZnO not only acts as the catalyst support but also acts as a catalyst for the acetone condensation reaction, especially at higher temperatures above 200 °C. On the other hand, at lower temperatures (as low as 100 °C), the presence of *n*-ZnO in the physical mixture leads to an increase in the yield of IPA. The reaction route was compared by studying changes in the reaction over time and evaluating different intermediates as starting materials, specifically, the selective hydrogenation of acetone over CAT-IV. This catalyst system provided some insight into

the efficient synthesis of IPA from acetone in high yields. However, a drawback of this catalyst was its primary composition of a hydrogenation catalyst, physically mixed with an acidic catalyst, and it was gradually reduced to the more metallic form of ruthenium. Supporting noble metals (hydrogenation catalyst) on the solid acid-base catalyst, which helps to maintain the catalyst in its metal oxide form, may improve the catalyst system.

## 3. Materials and Methods

### 3.1. Catalyst Preparation

*n*-ZnO was used as a support and was prepared as described previously by Bagabas and his co-workers [6]. As an illustration of the detailed synthesis, the zinc pyruvic acid oxime complex was synthesized from zinc sulphate heptahydrate, sodium pyruvate, hydroxylamine hydrochloride, and sodium bicarbonate. The complex, $Zn(PAO)_2(H_2O)_2$, was pyrolyzed to generate zinc oxide nanoparticles (*n*-ZnO). Ruthenium 5% on activated charcoal (*n*-Ru/AC) was commercially available and was used without further purification [24]. The *n*-ZnO with *n*-ruthenium-supported on activated charcoal (*n*-Ru/AC) was physically mixed and ground at different weight ratios. The mixture was pulverized and subsequently calcined at 400 °C for 12 h. The *n*-Ru/AC:*n*-ZnO ratios were 1:2 (CAT-II), 1:1 (CAT-III), and 3:2 (CAT-IV). For comparison purposes, pure *n*-ZnO (CAT-I) and pure *n*-Ru/AC (CAT-V) with 5.0% wt./wt. of Ru were studied. Some basic properties of the composite catalysts and the nominal ruthenium loadings are listed in Table 1.

### 3.2. Catalyst Characterization

XRPD patterns of the calcined catalysts were obtained by using a Philips X'pert pro diffractometer, operated at 40 kV and 40 mA by using $CuK_\alpha$ radiation in the 2-Theta range from 10° to 80° in steps of 0.02°, with a sampling time of 1.0 s/step. The average crystallite size was calculated by using Scherrer's equation. A Philips CM-200 FEG microscope, 200 kV, equipped with a field emission gun, was used for transmission electron microscopy (TEM) analysis.

The Brunauer, Emmett, and Teller (BET) specific surface area of the prepared catalysts was experimentally measured by $N_2$-physisorption at −196.0 °C by using a Quantachrome Autosorb-1-MP instrument. $H_2$-TPR experiments were conducted using a Quantachrome CHEMBET 3000. The experiments were carried out using a gas mixture of 5.0% $H_2$ in Ar at a flow rate of 30 mL min$^{-1}$ with a ramp rate of 10 °C min$^{-1}$ from 25 °C to 600 °C. The metal surface area and dispersion were determined by $H_2$ pulse chemisorption experiments by using a micrometrics AutoChem 2910 instrument, where 50.0 mg of catalyst was treated at 150 °C for 16 h under vacuum before starting the experiment. After completion of the reduction by hydrogen, the temperature of the sample was decreased to 70 °C, and pulses of $H_2$ were sent to determine the volume of $H_2$ uptake. The metal surface area and dispersion were determined from the hydrogen consumption by the sample up to full coverage saturation of the surface.

### 3.3. Catalyst Testing

Before the activity tests, the catalysts were activated by a reductive treatment at 400 °C under an $H_2$ gas flow of 60 mL·min$^{-1}$ overnight. The hydrogenation of acetone was performed at reaction temperatures of 100, 150, 200, 250, 300, and 350 °C in a glass tubular fixed-bed microreactor (ID 0.9 cm). The $H_2$/acetone mole ratio was varied at 1.5, 3.0, 4.5, or 6.0. Acetone was delivered to the reactor by a Hitachi HPLC pump (Model L-2100/2130). The acetone was first evaporated at 65 °C to avoid premature condensation and was mixed with pure hydrogen at the top of the catalyst bed. A K-type thermocouple was placed in the center of the catalyst bed to monitor the actual reaction temperature. The liquid products were condensed at −40 °C and were collected every hour, and subsequently were analyzed offline by using a Varian CP-3800 gas chromatograph, equipped with a flame ionization detector (FID) and a 50 m, 0.32 mm inner diameter, 1.2 μm thick CP-Wax 58 CB column. The gaseous effluent was analyzed online by a refinery gas analyzer Varian CP-3800 GC,

equipped with an FID and TCD. The products were identified by GC using standard samples. A simple schematic diagram of the experimental rig is shown in Figure 10.

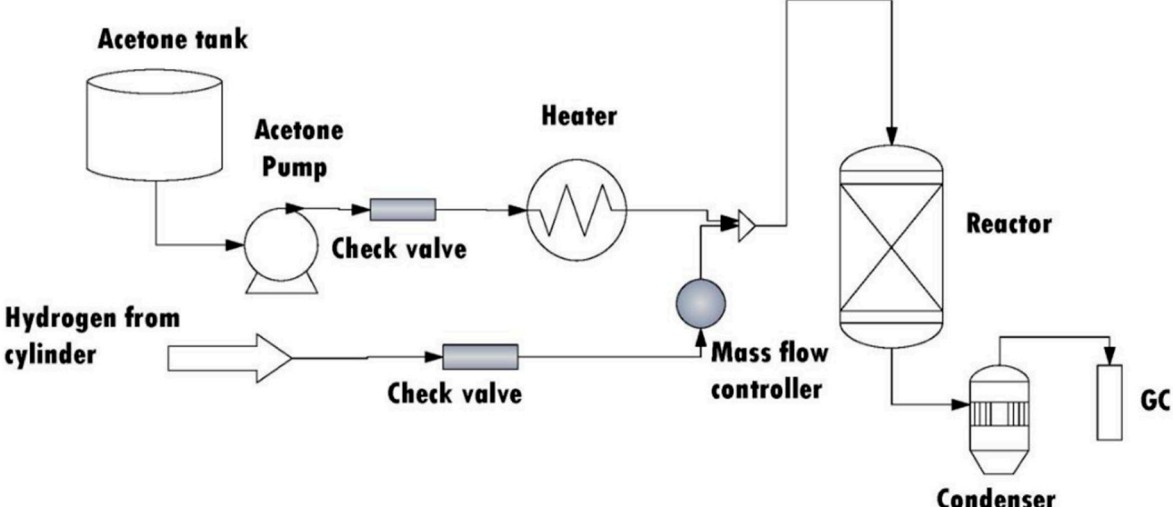

**Figure 10.** A simple schematic diagram of the experimental rig.

## 4. Conclusions

In conclusion, we demonstrated that under moderate conditions (100 °C, 0.1 MPa $H_2$), it was possible to selectively and quantitatively convert acetone to much valuable IPA (selectivity 99.8%) using a combination of an acid-base catalyst (to catalyze condensation) and a hydrogenation catalyst. The activity and selectivity of IPA in the hydrogenation of acetone, over Ru/C-ZnO composite catalysts depended critically on the Ru particle sizes and temperature.

Furthermore, an important feature of this reaction system is that ZnO not only acts as the catalyst support but also acts as a catalyst for the acetone condensation reaction, and thus, the presence of *n*-ZnO in the physical mixture, probably by maintaining ruthenium in metallic form, leads to an increase in the yield of IPA. The reaction route was compared by studying changes in the reaction over time and evaluating different intermediates as starting materials, specifically, the selective hydrogenation of acetone over CAT-IV. This catalyst system provided a novel and economical route to the efficient synthesis of IPA from acetone in high yields. However, a drawback of this catalyst is that it is primarily composed of a hydrogenation catalyst, physically mixed with an acidic catalyst, and it is gradually reduced to the more metallic form of ruthenium. Supporting noble metals (hydrogenation catalyst) on the solid acid-base catalyst, which helps to maintain the catalyst in its metal oxide form, may improve the catalyst system.

The deactivation of CAT-IV catalyst was not observed within the time span of five hours of time on stream, in which we believe that this increased stability was due to the good dispersion of ruthenium on activated carbon and the stabilization effect of *nano*-sized zinc oxide.

## 5. Patents

US 8110708, 7 February 2012 and US 8362302, 29 January 2013 patents are published as a part of this study.

**Author Contributions:** Conceptualization, A.A.A.-R. and A.A.B.; methodology, A.A.A.-R., V.M.A., M.M.M.M. and A.A.B.; formal analysis, A.A.A.-R. and A.A.B.; writing—original draft preparation, A.A.A.-R., I.B. and A.A.B.; project administration, A.A.B. All authors have read and agreed to the published version of the manuscript.

**Funding:** We thank King Abdulaziz City for Science and Technology (KACST) for funding this work under project No. 29-280.

**Data Availability Statement:** Not applicable.

**Acknowledgments:** We also thank King Saud University, Deanship of Scientific Research, College of Engineering Research Center for their support.

**Conflicts of Interest:** The authors declare no conflict of interest.

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
