# Peer review of "Highly Selective Gas-Phase Catalytic Hydrogenation of Acetone to Isopropyl Alcohol"

_catalysts, doi:10.3390/catal12101251_

Round 1
Reviewer 1 Report
Comments was uploaded as attached

Author Response
We would like to thank the reviewer for his insightful comments. We took all of them into consideration and rewrote the manuscript accordingly. Please find below our responses to your comments. Furthermore, we highlighted all the amendments in the manuscript for your perusal.
1st Comment:
It is hardly to see the scale (XX nm) in TEM Figure, as well as the particle size of Ru in the range of 3-4 nm.
Response: We have made the scale bar clearer and now more visible TEM pictures are embedded into the manuscript.
2nd Comment:
Some description are inaccurate, such as High temperature led to the reduction of ruthenium defect sites (Line 124 in page 5), what is ruthenium defect sites?
Response:
Thank you very much for the comment.
Line 124, page 5 reads:
"High temperatures led to the higher reduction of ruthenium defect sites, which are responsible for acetone conversion."
Now changed to:
High operating temperatures led to lower adsorption of acetone over ZnO support, which was the main cause of the lower conversion of acetone, as it was proven previously by the absence of adsorbed acetone above 698 K by the vibrational spectroscopy [13].
3rd Comment:
What is the reason for the conversion of acetone decreasing with H2/DMK mole ration, whereas IPA selectivity did not change?
Response: This effect is purely due to the kinetics of the reaction, and the following explanation is added to the manuscript:
If the adsorption processes of reactants (H2, acetone) and isopropanol product on the Ru metal catalyst surface competed to a certain extent, the initial hydrogenation rate of acetone would decrease gradually with an increase in the hydrogen partial pressure, corresponding to the excess surface coverage of hydrogen with respect to the acetone. At higher hydrogen partial pressure, the rate would decrease owing to the higher surface coverage of hydrogen in comparison to acetone, resulting in lower reaction rates.
4th Comment:
What is DMK, there is no annotation observed in the manuscript.
Response: DMK stands for dimethyl ketone, another acronym for acetone. We showed and made an appropriate explanation for this annotation in the revised manuscript.
5th Comment:
The langue should be improved carefully, and lots of minor errors should be avoided, such as “XRD and XRPD”, “H2” (Line 273, page 11), “CATT-”in line 109 on page 5.......
Response: We revised the English language and corrected the typographical errors thoroughly in the revised manuscript.
6th Comment:
What is the role of ZnO, since the highest IPA with the lower conversion of acetone was achieved on CAT-V, i.e., Ru/AC without ZnO. However, in the presence of ZnO (CAT-IV), the IPA selectivity and acetone conversion are both high at 75oC.
Response: High operating temperatures led to lower adsorption of acetone over ZnO support, which was the main cause of the lower conversion of acetone, as confirmed in a previous publication [13]. Moreover, the catalyst identity plays a key role in acetone conversion and in directing the reaction towards either MIBK or IPA. In heterogeneous catalytic carbonyl hydrogenation, a general mechanism, referred to as C=O activation, is provided by the electropositive species such as H+ [14], Ru4+ [15], Znx+[16] on the catalyst surface. In the C=O activation, these electropositive species can coordinate with the oxygen atom in the C=O bond and polarize the carbonyl group, facilitating the attack of C=O by the positively hydrogen species. The generation of dissociative hydrogen is a facile reaction on highly active metal surfaces, such as ruthenium, while selective, non-competitive adsorption of acetone on zinc oxide surfaces occurs below 698 K [13]. Therefore, we are required to have additional surfaces for the adsorption of acetone in competition with hydrogen.
The molecular adsorption of acetone occurs because the carbonyl oxygen atom contacts either to the metal surface [17] or, here in our catalyst, to nonpolar zinc oxide sites [13]. The dissociated hydrogen atoms on the metal surface then attack the adsorbed acetone, where hydrogenation of the enolated carbonyl-carbon atom first occurs via isopropoxy intermediates (partially hydrogenated intermediates), followed by consecutive hydrogenation of these intermediates to isopropanol.
7th Comment:
The description is conflicting in Line 196-201 on page 8, so which is better for IPA formation, the metallic Ru or Ru oxide? More characterization should be given, such as XPS, as well as NH3-TPD since the manuscript points out the acid-base catalyst in line 339 on page 13.......
Response: In our case, metallic ruthenium is responsible for the formation of IPA, while both ruthenium and ruthenium oxide are claimed for such task in the previous publication of other research group. Therefore, metallic ruthenium is the common sites for the formation of IPA in both of our current investigation and the other group investigations.
An acidity study for our catalysts was published previously in Catal. Lett.: https://doi.org/10.1007/s10562-014-1253-1.
8th Comment:
“The activity and selectivity of IPA...depended critically on the Ru particle sizes” was concluded in conclusions, but no direct proof was shown to support this point.
Response: We mistakenly wrote Ru particle size rather than Ru loading, as we discussed in the manuscript. We fixed this unintended mistake in the revised version of our manuscript and correlate the activity and selectivity to Ru loading weight percentage.
Reviewer 2 Report
The manuscript “Highly Selective Gas-phase Catalytic Hydrogenation of Acetone to Isopropyl Alcohol” devoted to the direct selective hydrogenation of acetone over Ru/AC catalysts modified with ZnO. The topic of the research is not of high novelty. The manuscript in its current form cannot be accepted for publication. The main comments and questions are listed below:
1. Author claimed that the catalyst showed high activity at 75oC and the conversion reached almost 99% (Table 2). However, in the manuscript there are no data for the activity and selectivity of the catalyst at 75oC. The experimental curves for the conversion and selectivity started from 375K (i.e. 375-273=102oC). Therefore, the kinetic study is also questionable. Why the Authors decided that hydrogen dissociation is a rate limiting step?
2. If the above-mentioned conditions (75oC, H2/acetone=4.5) were chosen as optimal, why other experiments (effect of space velocity, effect of time on stream) were provided at 250 oC. Moreover, the results obtained at the same conditions are contradictory to these depicted in Fig. 3.
3. There is no information about the reaction conditions used for the comparison of the activity of different catalysts.
4. What was the role of zinc oxide?
The manuscript is not ready for publication.
Author Response
We would like to thank the reviewer for his insightful comments. We took all of them into consideration and rewrote the manuscript accordingly. Please find below our responses to your comments. Furthermore, we highlighted all the amendments in the manuscript for your perusal.
The manuscript “Highly Selective Gas-phase Catalytic Hydrogenation of Acetone to Isopropyl Alcohol” devoted to the direct selective hydrogenation of acetone over Ru/AC catalysts modified with ZnO. The topic of the research is not of high novelty. The manuscript in its current form cannot be accepted for publication. The main comments and questions are listed below:
Comments 1: Author claimed that the catalyst showed high activity at 75oC and the conversion reached almost 99% (Table 2). However, in the manuscript there are no data for the activity and selectivity of the catalyst at 75oC. The experimental curves for the conversion and selectivity started from 375K (i.e. 375-273=102oC). Therefore, the kinetic study is also questionable. Why the Authors decided that hydrogen dissociation is a rate limiting step?
Response: As the reviewer is absolutely right in his comment No. 1, in the processing of the data in Figure 3, we did not include the obtained data at 75oC. Thus, we removed all the references and points given for 75oC. And we left Figure 3 intact since the modelling data was based on the data used to draw this Figure 3 and others too. The kinetic model is not questionable because of this point, where we tried and utilized several proposed models of others and selected this purely on the basis of the quality of observation for a wide range of experimentation and goodness of fit.
Comments 2: If the above-mentioned conditions (75oC, H2/acetone = 4.5) were chosen as optimal, why other experiments (effect of space velocity, effect of time on stream) were provided at 250 oC. Moreover, the results obtained at the same conditions are contradictory to these depicted in Fig. 3.
Response: We changed all the references to 75oC optimal conditions to a new value of 100oC in line with Figure 3. Although optimal conditions were below these conditions, we purposefully chose this relatively high temperature to make changes more visible and striking. At around the optimal conditions, changes in the observed parameters were as such not striking.
Comments 3: There is no information about the reaction conditions used for the comparison of the activity of different catalysts.
Response: We forgot to include the reaction conditions as below (page 9, line 229): “Reaction conditions: 0.25 g catalyst, t = 250°C, H2/acetone mole ratio = 6:1.”
Comments 4: What was the role of zinc oxide?
Response: The comment has been considered. The following paragraph was added to the revised manuscript (please check page 5, line 145)
High operating temperatures led to lower adsorption of acetone over ZnO support, which was the main cause of the lower conversion of acetone, as it was proven previously by the absence of adsorbed acetone above 698 K by the vibrational spectroscopy [13]. Moreover, the catalyst identity plays a key role in acetone conversion and in directing the reaction towards either MIBK or IPA. In heterogeneous catalytic carbonyl hydrogenation, a general mechanism, referred to as C=O activation, is provided by the electropositive species such as H+ [14], Ru4+ [15], Znx+[16] on the catalyst surface. In the C=O activation, these electro-positive species can coordinate to the oxygen atom in the C=O bond and polarize the car-bonyl group, facilitating the attack of C=O by the positively hydrogen species. The genera-tion of dissociative hydrogen is a facile reaction on highly active metal surfaces, such as ruthenium, while selective, non-competitive adsorption of acetone on zinc oxide surfaces occurs below 698 K [13]. Therefore, we are required to have additional surfaces for the ad-sorption of acetone in competition with hydrogen.
The molecular adsorption of acetone occurs because the carbonyl oxygen atom con-tacts either to the metal surface [17] or, here in our catalyst, to nonpolar zinc oxide sites [13]. The dissociated hydrogen atoms on the metal surface then attack the adsorbed ace-tone, where hydrogenation of the enolated carbonyl-carbon atom first occurs via iso-propoxy intermediates (partially hydrogenated intermediates), followed by consecutive hydrogenation of these intermediates to isopropanol.
Reviewer 3 Report
The work “Highly Selective Gas-phase Catalytic Hydrogenation of Acetone to Isopropyl Alcohol” deals with a topic of high interest, and the results obtained are interesting, especially considering that they are obtained in a flow reactor. However, it is advisable to carry out a thorough review of the manuscript before its acceptance. The introduction should be expanded, it is proposed to make a re-organization of the presentation of the results to be able to read the work in a logical sequence. On the other hand, it is advisable to improve the discussion of the results that correlate the characterization of the catalysts with activity/selectivity. Probably, the following points are some of the points to address to improve the work:
The introduction needs to be improved. For example, the introduction does not clarify why catalysts containing ZnO are used in their formulation. New references must be included.
The technique used for the preparation of the Ru/AC catalyst is not described in the work. The authors could include it in the experimental section.
Could the authors include a schematic of the flow reaction system? At what pressure are the measurements made? What is the flow rate and mass of catalyst used in each experience?
In Table 1, which species present the assigned crystalline size? Although ZnO is mentioned in the text (line 93), the table is in another section and should be clarified.
The quality of the XRD diagram figure should be improved. Could the authors include the DRX profile of the other catalysts?
It has been reported that the acetone hydrogenation reaction is sensitive to the crystal size of Ru. Could the authors include the data on the sizes of the metallic particles of catalysts II, III, and IV?
Could the authors calculate the metallic dispersion? They have measurements of the metallic area.
Why is Catalyst V chosen to perform activity measurements as a function of temperature? Couldn't you put figure 6 before? In addition, in line 126 results of catalysts are mentioned that have not yet been shown.
Regarding the comment on acidity in line 127, could the authors indicate how the acid-base properties of the catalysts could influence the activity? Several sentences are included that mention the influence of the acid-base properties, but it is not clear in the work how this property affects the activity and selectivity of the catalysts.
How does metal dispersion affect activity? Is there any correlation between conversion/selectivity with dispersion?
Author Response
We would like to thank the reviewer for his insightful comments. We took all of them into consideration and rewrote the manuscript accordingly. Please find below our responses to your comments. Furthermore, we highlighted all the amendments in the manuscript for your perusal.
The work “Highly Selective Gas-phase Catalytic Hydrogenation of Acetone to Isopropyl Alcohol” deals with a topic of high interest, and the results obtained are interesting, especially considering that they are obtained in a flow reactor. However, it is advisable to carry out a thorough review of the manuscript before its acceptance. The introduction should be expanded, it is proposed to make a re-organization of the presentation of the results to be able to read the work in a logical sequence. On the other hand, it is advisable to improve the discussion of the results that correlate the characterization of the catalysts with activity/selectivity. Probably, the following points are some of the points to address to improve the work:
Comment 1: The introduction needs to be improved. For example, the introduction does not clarify why catalysts containing ZnO are used in their formulation. New references must be included.
Response: The following paragraph was included in the introduction;
As is well known in this art of using metallic catalysts, the catalytic performance depends on various factors such as nanoparticle size, nature of support, ease of regeneration, and reaction conditions [4]. The prior art has previously suggested that ruthenium hydrogenation catalysts could be reactivated by reducing the catalyst. Besides SiO2 [12], other non-reducible supports such as CNT [13], and MgO [14] were used as supports of ruthenium and were found to contribute to changes in the catalytic activity and selectivity in the hydrogenation of acetone and aliphatic aldehydes. Because the selective hydrogenation of aldehydes and ketones is known to be a reaction sensitive to metal– support interactions, it was stimulating to test the properties of Pt/ZnO catalysts for this reaction, knowing that ZnO showed promoting effects in several studies. A marked decrease in the catalytic activity with a significant increase in selectivity was observed in butadiene hydrogenation to olefins when the Pd/ZnO catalyst [15] was reduced above 423 K. In this context, our paper is devoted to the hydrogenation of acetone in the gas-phase, at ambient pressure, on Ru deposited on activated charcoal and composited with ZnO to improve the adsorption of acetone, under various operating conditions of temperature, flow rate, and H2/acetone mole ratio. Thus, we aimed to combine the good hydrogenation ability of ruthenium with the adsorption capabilities of ZnO.
Altogether, 16 new relevant references are added to the manuscript.
Comment 2: The technique used for the preparation of the Ru/AC catalyst is not described in the work. The authors could include it in the experimental section.
Response: We added a new paragraph explaining the details of nRu/AC (page 12, line 347):
Ruthenium 5% on activated charcoal (n-Ru/AC) was commercially available and used without further purification [31].
Comment 3: Could the authors include a schematic of the flow reaction system? At what pressure are the measurements made? What is the flow rate and mass of catalyst used in each experience?
Response: The comment was considered. A schematic diagram of the experimental rig was added (please check page 13, Figure 10).
The reaction was performed at ambient pressure with an amount of 0.25 g catalyst.
Comment 4: In Table 1, which species present the assigned crystalline size? Although ZnO is mentioned in the text (line 93), the table is in another section and should be clarified.
Response: The crystallite sizes in Table 1 are for zinc oxide because it was the only detected phase in the XRD patterns.
Comment 5: The quality of the XRD diagram figure should be improved. Could the authors include the DRX profile of the other catalysts?
Response: We included a new XRD diagram with excellent quality, showing the patterns of all catalysts.
Comment 6: It has been reported that the acetone hydrogenation reaction is sensitive to the crystal size of Ru. Could the authors include the data on the sizes of the metallic particles of catalysts II, III, and IV?
Response: We mistakenly wrote Ru particle size rather than Ru loading, as we discussed in the manuscript. We fixed this unintended mistake in the revised version of our manuscript and correlated the activity and selectivity to Ru loading weight percentage.
Comment 7: Could the authors calculate the metallic dispersion? They have measurements of the metallic area.
Response: The dispersion of ruthenium was calculated in our previous paper, published in Catal. Lett.: https://doi.org/10.1007/s10562-014-1253-1.
Comment 8: Why is Catalyst V chosen to perform activity measurements as a function of temperature? Couldn't you put figure 6 before? In addition, in line 126 results of catalysts are mentioned that have not yet been shown.
Response: We did choose CAT-IV, not CAT-V, to investigate the impact of temperature on the catalytic activity because CAT-IV was the best catalyst for the direct hydrogenation of acetone to IPA. Please re-check what we wrote in lines 123–126.
Comment 9: Regarding the comment on acidity in line 127, could the authors indicate how the acid-base properties of the catalysts could influence the activity? Several sentences are included that mention the influence of the acid-base properties, but it is not clear in the work how this property affects the activity and selectivity of the catalysts.
Response: Please refer to our previous study, published in Catal. Lett. Because we did not do any new ammonia or carbon dioxide based TPD studies, it was a mistake to mention acid-base properties, as if it has been done in this study. Thus, we removed all the discussions pertaining to acid base characteristics.
Comment 10: How does metal dispersion affect activity? Is there any correlation between conversion/selectivity with dispersion?
Response: Thank you for the questions. The explinations for the questions were discussed in our previous study published in Catal. Lett.: https://doi.org/10.1007/s10562-014-1253-1.
Round 2
Reviewer 1 Report
The authors clarified the questions raised, so it is OK to publised now.
Author Response
We sincerely appreciate the time and efforts of the reviewer. The thoughtful comments helped us improve the manuscript.
Reviewer 3 Report
I have carefully read the new version of the work and the authors' responses. Part of the doubts have been clarified, but there are still some pending points to improve so that the work can be accepted.
1- In some figures, the temperature is indicated in K and others in °C. In the discussion of results, the results are expressed in °C. Please homogenize units.
2- Some characterization data (Figure 1, Table 1) have already been previously published by the authors in "Catal Lett DOI 10.1007/s10562-014-1253-1". (Ref. 6 in this work). Please indicate that they have already been published and cite them appropriately when they appear in this new manuscript.
3- Why the effect of the hydrogen to acetone mole ratio was studied at 75°C? This temperature is not part of the analysis performed in section 2.1. Please include in Figure 3 some results obtained at 75°C, to have some common parameters-
4-When the Effect of space velocity is studied, does the H2/DMK ratio vary? since the flow rate of acetone remains constant. The catalyst mass also remains constant? This point needs to be clarified. Also, please explain why it was done at 250°C.
5- Since you already have dispersion values ​​calculated and published in a previous job (for another reaction), please indicate if there is any correlation between dispersion and activity for the reaction studied in this work.
Author Response
We would like to take this opportunity to thank the reviewer for the thoughtful comments on this manuscript and for the insightful points to improve it. The manuscript has been revised as per the comments given by the reviewer, and our responses to each comment are shown below.
1- In some figures, the temperature is indicated in K and others in °C. In the discussion of results, the results are expressed in °C. Please homogenize units.
Response to comment 1: As you have appropriately commented on the homogenization of units of temperature, we have now converted all to the C rather than the K.
2- Some characterization data (Figure 1, Table 1) have already been previously published by the authors in "Catal Lett DOI 10.1007/s10562-014-1253-1". (Ref. 6 in this work). Please indicate that they have already been published and cite them appropriately when they appear in this new manuscript.
Response to comment 2: The comment has been considered. We have included the reference "Catal Lett DOI 10.1007/s10562-014-1253-1" (Ref. 6 in this work) in Figure 1 and Table 1, as shown in the manuscript.
3- Why the effect of the hydrogen to acetone mole ratio was studied at 75°C? This temperature is not part of the analysis performed in section 2.1. Please include in Figure 3 some results obtained at 75°C, to have some common parameters-
Response to comment 3: Yes we have tested at 75 C as well, but we have not included it in Figure 3 for two reasons; first one to make the graph more even spaced and good looking and the second reason was there was no appreciable difference between 75 and 100 C, hence data obtained at 75 C was not included in the Figure 3.
4-When the Effect of space velocity is studied, does the H2/DMK ratio vary? since the flow rate of acetone remains constant. The catalyst mass also remains constant? This point needs to be clarified. Also, please explain why it was done at 250°C.
Response to comment 4: We have on-purpose studied at 250 C, since at the high temperature there is a competing reaction of using acetone to produce Methyl Isobutyl ketone (MIBK), at these temperatures, a meaningful comparison of catalytic activity of various catalysts with a view to improve selectivity to IPA was only possible at this high temperature.
5- Since you already have dispersion values ​​calculated and published in a previous job (for another reaction), please indicate if there is any correlation between dispersion and activity for the reaction studied in this work.
Response to comment 5: We would like to thank the reviewer for the question. As is well known in metallic catalysts, the degree of dispersion of the active constituent affects the catalytic activity of the heterogeneous catalysts.
However, in our present study, we did not include metallic surface areas and dispersions since the reaction studied here is not purely related to the properties of metal surfaces, but also related to zinc oxide of which contribution from active support, namely zinc oxide was not quantified. Hence we have neither included the data on metallic properties nor speculated on the metallic contribution to the activity or selectivity.
